# Safety Performance in Acute Medical Care: A Qualitative, Explorative Study on the Perspectives of Healthcare Professionals

**DOI:** 10.3390/healthcare9111543

**Published:** 2021-11-12

**Authors:** Lina Heier, Donia Riouchi, Judith Hammerschmidt, Nikoloz Gambashidze, Andreas Kocks, Nicole Ernstmann

**Affiliations:** 1Institute for Patient Safety (IfPS), University Hospital Bonn, 53127 Bonn, Germany; do.riouchi@gmail.com (D.R.); judith.hammerschmidt@ukbonn.de (J.H.); nikoloz.gambashidze@ukbonn.de (N.G.); nicole.ernstmann@ukbonn.de (N.E.); 2Center for Health Communication and Health Services Research (CHSR), Department for Psychosomatic Medicine and Psychotherapy, University Hospital Bonn, 53127 Bonn, Germany; 3Directorate of Nursing, University Hospital Bonn, 53127 Bonn, Germany; andreas.kocks@ukbonn.de

**Keywords:** patient safety, occupational safety, safety performance, healthcare professionals, nursing, acute care, qualitative research

## Abstract

Healthcare professionals need specific safety performance skills in order to maintain and improve patient safety. The purpose of this study is to get a deeper understanding of healthcare professionals’ perspective in acute care on the topic of safety performance. This study was conducted using a qualitative approach. Healthcare professionals working in nursing were interviewed using semi-structured interviews. Using content analyzing, categories were identified which present aspects of safety performance; subcategories were developed deductively. A total of 23 healthcare professionals were interviewed, of which 15 were registered nurses, five were nursing students and three were pedagogical personnel. Nine (39.1%) were <30 years old, 17 (73.9%) were female, and 9 (39.1%) had a leadership function. Results highlight the importance of safety performance as a construct of occupational health rather than of patient safety, and the role of the organization, as well as the self-responsibility of healthcare professionals. Healthcare professionals should be more conscious of their role, have a deeper understanding of the interaction of individual, team, patient, organization and work environment factors.

## 1. Introduction

With an occurrence of 8 to 12% of all hospitalizations in European countries, adverse events have a significant impact on patient outcomes [1]. Hospitalized patient outcomes, such as mortality, hospital-acquired pneumonia, catheter-associated urinary tract infection and pressure sores, are directly associated with nurse-to-patient ratio, training and staffing, and work experience, among other factors [2,3,4]. Patient safety and health do, therefore, directly depend on healthcare professionals (HCP), especially nurses skills, knowledge and well-being [5,6,7]. The nurses’ safety, well-being and safe care of patients are related to nurses’ working environment [2,3,5,7,8,9,10,11]. The National Academy of Science identified in nurses’ work and work environments several aspects which are evolving over time and influencing patient safety in a clinical setting: more complex, multimorbid clinical conditions of patients, shorter hospital stays, redesigned work, changes in the deployment of nursing personnel, frequent patient turnover, high staff turnover, long work hours, a rapid increase in new knowledge and technology and increased interruptions and demands [9].

These factors indicate that organizational and technical aspects, along with team and individual elements, affect patient safety. The human factors approach aims to improve patient safety by questioning and establishing how systems work and how this complexity affects patient safety [12,13]. Human factors and ergonomics are scientific disciplines that aim to produce knowledge to redesign and improve processes [12,13,14]. Human factors refer to environmental, organizational and job factors, and human and individual characteristics that influence behavior at work [15]. It follows that organizational factors will affect patient safety, but the team and individual aspects will equally influence the behavior of nurses and other HCPs concerning safe patient care [12,13,16].

As a construct consisting of safety participation and safety compliance, HCPs’ safety performance plays a key role in providing safe care, consequently maintaining and improving patient safety [17]. The term safety compliance is used to describe the core activities that need to be carried out by individuals to maintain workplace safety [17]. These behaviors include adhering to standard work procedures and wearing personal protective equipment. The term safety participation is used to describe behaviors that do not directly contribute to individual safety but help develop an environment that supports safety [17,18].

The association of HCP behavior and patient safety has been thoroughly studied, using a quantitative or mixed-method approach [2,4,19,20,21,22,23,24,25,26,27]. HCPs who work in nursing, and their unique views on safety performance regarding Griffin and Neals conceptualization [18], their role, and expectations for their work environment in acute medical care in Germany are rare. This study aims to explore HCP perspectives on the topic of safety performance with a qualitative approach.

## 2. Materials and Methods

### 2.1. Study Design

This qualitative interview study is part of the explorative mixed-methods SPOHC study (Safety Performance of Healthcare Professionals), conducted in 2018–2020. The study received ethical approval from a local ethics committee in Germany (number 075/19). SPOHC is built upon the integrative workplace safety model and focuses on safety performance as a construct of safety compliance and safety participation [28,29]. The SPOHC data collection methods comprised qualitative interviews and a cross-sectional written survey with healthcare professionals. The SPOHC survey results focus on the testing and validation of two instruments (a workplace health and safety instrument and situational judgement test) to measure the safety performance of HCP in Germany. Both instruments show acceptable psychometric properties, allowing new possibilities to measure the construct of safety performance [24,30].

### 2.2. Sample and Study Setting 

The sample was based on convenience sampling and consisted of registered nurses, nursing students (last year of training) and pedagogical personnel working in nursing in one university hospital, two university teaching hospitals and two nursing schools. Registered nurses in Germany generally undergo a three-year training program integrated into nursing schools with a state examination. University qualifications in nursing, which are standard internationally, have only a short tradition in Germany and, so far, account for only a small proportion of about one to two percent of the nursing teams in hospitals [31]. Nursing schools are traditionally part of hospitals; consequently, nursing students work on the frontline from the beginning of the training program, attended by their supervisors. The focus of their work is to assist patients with physical care, assist team members, provide guidance and supervision to patients and their families. In some long-term psychiatric departments, staff with a pedagogical education are also part of the multiprofessional nursing team. They take on nursing-therapeutic tasks, especially in areas of child and adolescent psychiatry, and care for patients in these contexts. Nursing-therapeutic tasks can be e.g., developing the structure of the day or monitoring of the patient in working groups. The multi-professional nursing team can therefore consist of registered nurses, nursing students and pedagogical personnel to ensure high quality care on several levels. Nurses who have completed a one or two-year training program to be a nursing assistant were excluded from the study.

Nursing managers and headmasters of nursing schools were informed about the study via email and personal contact. The SPOHC project was presented during regular team meetings by a researcher with a clinical and nursing science background, and questions regarding goals, data protection, process, and effort could be answered directly. All HCPs were precisely informed about the protection of their person and data as well as the publication of the results. It was ensured that participation was completely anonymous and that no conclusions could be drawn about individuals or teams. If the HCP expressed interest in participating, they were subsequently contacted by email, with data protection documents and consent forms. Subsequently, with the HCP’s consent, an appointment was made for the interview.

### 2.3. Data Collection

Two female researchers with a nursing science background, and a female student assistant with a psychology background, conducted semi-structured, face to face interviews with HCPs who were working in nursing between July 2019 and March 2020. Both researchers and the student assistant are trained in qualitative data collection and data analysis topics.

The semi-structured interview content was developed with the CRSS method to develop interview guidelines: C = collect, R = review, S = sorting, S = summarize [32]. The first step was a brainstorming process to collect questions, followed by a review step to sort out all closed, evaluative, and suggestive questions [32]. In the next step, questions were sorted by content and in the last step, summarized [32]. The brainstorming process and first collection of questions in step one was influenced by own prior clinical experience, publications on safety performance, and the theoretical model (the integrative workplace safety model) on which the overall SPOHC study is based [28,29].

The guidelines consisted of four key questions regarding aspects and barriers of safety performance, the own role and enhancements for work on the frontline (detailed information about the key questions is presented in Figure 1). The key questions were designed to achieve descriptions of specific situations and procedures at the frontline to explore realistic situations and let the participant reflect on their performance and role as a HCP. The semi-structured interview was pre-tested with a study nurse working in health services research and with clinical experience.

All interviews were conducted at the workplace in separate rooms without any interruptions. At the beginning of the interview, the researcher introduced herself and explained their clinical background to establish a trustworthy situation. HCPs were informed about voluntary participation, data protection, the possibility of termination at any time, and the study’s aim. Sociodemographic information was collected at the end of the interviews.

### 2.4. Data Analysis

Each interview was audio-recorded, fully transcribed, pseudonymized and coded using content analysis. Categories were developed deductively, main categories were identified from the guideline, subcategories were based on the human factors model of patient safety [12]. The four main categories which have been identified as the most relevant for patient safety were (1) Organizational/Managerial; (2) Workgroup/Team; (3) Individual Worker; (4) Work environment [12]. We used these categories and an additional category (5) Patient/Caregiver as the subcategories in the performed content analysis. One female researcher with a background in nursing science and clinical nursing, and a female student assistant with a psychology background, who both were responsible for the data collection coded the interview transcripts independently and discussed all text segments and codes using the software MAXQDA (version 18/20, VERBI GmbH, Berlin, Germany). Afterwards, the text segments were paraphrased, generalized, and reduced, based on the content analysis recommendation form of content structuring of Mayring [33]. All anchor quotes were translated into English by a translation agency. All findings were discussed by a multidisciplinary team of researchers working in patient safety with a background in health services research, nursing science, and psychology. The transcripts or results of data analysis were not discussed with the interview partners themselves.

### 2.5. Trustworthiness of the Study

To ensure credibility, transferability, dependability and confirmability, our study is built upon the framework presented by Korstjens and Moser [34]. To ensure credibility, investigator triangulation was used, and two researchers coded, analyzed and interpreted the data. To ensure transferability, we sought to provide thick descriptions of context, as well as behavior and experiences. To ensure dependability and confirmability, we endeavored to report the different qualitative research steps we conducted in a transparent manner.

## 3. Results

### 3.1. Sample Characteristics

Fifteen registered nurses, five nursing students and three pedagogical personnel, all working in nursing, were interviewed. From these 23 interview partners, nine (39.1%) were <30 years old, 17 (73.9%) were female, and nine (39.1%) had a leadership function. A total of 15 (65.2%) had worked longer than five years in nursing, and 13 (56.5%) had worked longer than five years in the same department.

### 3.2. Aspects of Safety Performance

In general, the interviewed HCP understood the general aspects of safety performance to mean everyday behavior related to the safety of patients, their family members and hospital staff. The focus was mainly on reducing risk factors and observing occupational health and safety, observing protective measures and theories on the occurrence of errors, which are typical, practical examples of accident prevention.


*Well, no idea, that there are no power cables on the floor that you can trip over.*

*(IP01)*



*Well, for example, that you, when you’ve moved a patient from one room to another, that you then lower the bed again, that you, I don’t know, also explain to the patient how the nurse call button works, adjust the lights, that, if it’s dark, you might also turn on the light and explain to the patient how to turn on the light.*

*(IP05)*



*It’s also about sharps disposal, correct waste disposal, avoiding situations that are potentially dangerous for patients, right?*

*(IP08)*



*Well, let’s start with patient safety; so, there, I would say that, for example, when the floor is mopped, that some sign is placed stating that like/that the floor is wet.*

*(IP14)*



*We had a construction site here a little while ago. So we had to be careful, too; there was scaffolding here on the patio, so we locked the patio door to make sure that patients do not go there and possibly climb up.*

*(IP19)*


### 3.3. Aspects of Safety Performance—Organizational/Managerial

In the interviews, concerning aspects of safety performance that address the organizational level, establishing rules and checking them was a particular focus. It was reported that management specifications, assessments, standards, and guidelines influence HCP safety performance. This also includes the mandatory use of Critical Incident Reporting Systems, checklists, patient wristbands and other instruments to increase patient safety. It should be emphasized that the organization’s rules should be reviewed by management to ensure consistent compliance.


*I believe at our facility, it’s that our director and deputy director are both people who pay very close attention to that. And if any mistakes are made, they communicate that. And they have very high quality standards for our team. And that as a result, I believe, a lot is actually achieved/that, well, people do act properly because we know that this is kind of demanded and required from us.*

*(IP10)*


In the interviews, HCPs emphasized that the organization offers regular training programs and that all HCPs (e.g., physicians) are required to attend the training courses.


*I believe, that is really because the people all receive really good initial training, a good briefing, and continued education. So, it’s not like someone just says: “Come on, let me show you the emergency kit really quickly”, but there is an actual continued education event where you sit down for two hours and where each drug is discussed, too, what its indication is and when to use it.*

*(IP07)*


From the interviewees’ point of view, the organization’s responsibility to provide a safe workplace is important. This includes good personnel key, personnel with sufficient language skills and the establishment of appropriate structures and processes, like emergency call systems, occupational safety committees, mandatory meetings after adverse events, monitoring of patient data protection, etc.

### 3.4. Aspects of Safety Performance—Team

Based on the interviews, HCPs emphasized the importance of sharing knowledge about patient safety within the team. The knowledge from training programs should be passed on in teams, managers must pass on their knowledge to their employees about safety issues and current measures; there should be regular team-internal meetings focusing on patient safety.


*That is, you then also have to take a second step and not only inform people but to somehow also enable them to act accordingly. And typically, this is best done by, well, by showing them how to do it.*

*(IP11)*


The interviewed HCPs reported that teamwork in the inter-professional and nursing team is characterized by responsibility, openness towards mistakes, and safety. It is about agreements, open communication, trust and the perception of problems and uncertainties of colleagues. The cooperation between different professions should be reflected upon, and ambiguities should be addressed and solved through supervision. From the HCPs’ point of view, teamwork described as good and relaxed promotes patient safety and safety culture within the team.


*That is, if we had a reanimation here last time, then the medical team was brought in, then I asked that we reflect again on what we did, how we did it, how everyone experienced it, how everyone felt in the process, and we try to reflect on that again on the larger scale and simply do better in the future to simply ensure patient safety that way as well.*

*(IP06)*


According to the participants, a team assumes a control function to detect errors early, familiarize new colleagues, and enable trustful teamwork.


*And we actually train our physicians a little bit in this way because: “Well, do this, do that”, no physician order. Now, we don’t do anything without a physician order. And sometimes, many physicians actually then try to verbally delegate things somehow, but we just don’t do it. And then they got used to it.*

*(IP03)*


### 3.5. Aspects of Safety Performance—Individual Worker

On an individual level, adherence to safety-related rules played a major role for the interviewed HCPs. The correct wearing of protective and work clothing was mentioned here; being informed about current safety-relevant standard operating procedures and measures, carrying out room checks, protecting patients from falling, observing hygiene rules and confidentiality, and working in a de-escalating manner.


*And most of it, well, it’s very important that personal protection, that it is always paramount. Because, if I’m down sick, I can no longer help others. That’s why I always start with myself.*

*(IP17)*


Based on the interviews, HCPs should be aware of their function, have a role model status, be responsible for transferring knowledge, seek inter-professional help in case of uncertainties, and admit mistakes. It involves keeping agreements, making routine situations safe, developing an awareness of dangerous situations.


*And especially the last case, it just showed me that even I, with twenty years of job experience, still need to always reflect. Work on myself. And that gave me a little more security, to still feel that. If I had gone in there indifferently and came out indifferently, I would have been rather worried, or actually, probably not.*

*(IP17)*


HCPs should be responsible for participating in further training programs, continually expanding their knowledge, and passing it on.


*And that you, as I said, participate in continued education, if you learn something from the continued education, that you just pass that on in the team, too.*

*(IP15)*


### 3.6. Aspects of Safety Performance—Work Environment

On the work environment level, structural measures such as clearly arranged departments, escape routes, emergency doors, fire alarms, alarm systems, and safe windows and doors have been mentioned as aspects that influence safety performance. Medical products such as bed rails, alarm mattresses and protective equipment for nursing professionals, as well as training courses on technical aspects of everyday work (digitalization in nursing), were mentioned by the HCP here.


*An example is, well, if a patient is infectious and isolated, you have to put on specific protective clothing if you perform activities near the patient so that you then leave the microbes in the room when you take off the protective clothing.*

*(IP11)*


The correct handling of medication by HCP was also mentioned. Here, hygienic aspects played a role as well as control mechanisms, storage systems and the placing and administration of these.


*Another topic is the administration of medications; for example, infusions, when I administer them. Or injections that I administer. There as well, it’s important that I make sure, for instance, to disinfect the puncture site, or disinfect the connectors to which the infusion is hooked up to ensure that I do not expose the patient to microbes through the injections or infusions.*

*(IP11)*


### 3.7. Aspects of Safety Performance—Patient

At the patient level, the aim is to avert dangerous situations and adverse events. Measures to protect patients must be initiated at an early stage; patients must be closely monitored to be protected immediately in case of risks—for example, the WHO checklist for avoiding adverse events during surgical procedures.


*For instance, storage, repositioning, patient admission, to ensure that data are appropriately collected, documented, and that this is a continuous cycle. The patient, for example, which side is operated on, is it the right patient, is the name correct, the information, etc.? Is the patient placed on the correct table? Have we brought up the correct X-rays? It runs through all of that. Well, those are the patient-relevant data that, I think, do play a major role. Because mix-ups have been described over and over. And of course, they should be avoided if at all possible.*

*(IP22)*



*So, of course, as I already mentioned, with regard to hazardous objects, escape routes, that patients have been informed, too, for instance, what to do in case of fire. Because something like that can happen at any time even without external influences.*

*(IP13)*


HCPs act as patient advocates; they are mainly responsible for patient safety. This includes providing support when uncertainties arise, providing information and assistance in decision-making, and communicating patience and time so that the patient feels supported, understood, and safe.


*My staff knows exactly, if I’m not well, that I simply know I can always address that. And to give the patient this psychological, well, safety; I do think that is part of patient safety as well.*

*(IP06)*


## 4. Discussion

Our qualitative study aimed to explore the perspectives of HCPs who are working in nursing in acute medical care on the topic of safety performance. Categories were developed deductively based on the human factors model of patient safety, and represent aspects of safety performance experienced by HCPs at the frontline [12,13]. Results highlight the importance of safety performance as a construct of occupational health rather than of patient safety and the role of the organization. The interviewed HCPs struggled to describe what safety performance means individually, and situations related to safety performance or general safety issues. The focus of interview participants was more on occupational safety aspects (for example, handling injection needles or technical handling of medication) or organizational or management aspects, than patient- or team-related aspects of safety. Safety performance was described as a functional construct of occupational health, e.g., to ensure that patients do not fall, the work environment is secured, or work clothes are worn. It involves factual information about aspects of occupational health and safety. The interview partners were asked to describe their experience with safe situations. The HCPs stated that, beyond the functional safety performance, factors regarding teamwork, communicational skills and responsibility aspects can also be classified as a level of interactive safety performance. Their roles and responsibilities regarding patient safety became clearer and structured while talking about their perspective on safe and unsafe situations.

However, it became apparent that one’s safety performance and role as a HCP in the hospital system were only superficially reflected upon, and the organization and management were described as playing a more important role. The organization should establish rules for constant compliance with high safety standards (e.g., using critical incident reporting systems, standardized handovers, safety rounds and speak up initiatives [35,36]) and checking them was a particular focus for the HCPs who work in nursing. Rules, checklists, and standards for nursing and physicians must be more strictly observed and verified by the management to improve safety performance. This is contrary to a previous study which found that nurses with higher autonomy by the organization also made fewer medication errors, and that this aspect was the only structural aspect related to patient safety [37]. The authors of this study attribute this effect to nurses’ higher education. The higher the qualification, the higher the autonomy, and the rarer the patient safety errors [37]. Other studies underline the correlation between safety performance and job autonomy as well [38,39,40]. Registered nurses in Germany are typically trained for three years, but not on university/college level, which is the international standard for becoming a registered nurse [31]. A 2015 survey found that 1% of all nurses in Germany who work in direct patient care have a college degree [31]. Future studies should clarify whether curricula differences in terms of safety performance between college and vocational training might contribute to the need for more monitoring management. One aspect of safety performance in all subcategories is the implementation and participation of training and qualification programs that address patient safety topics in education, training, continuing education, or degree programs for HCPs. The organizational offer of regular training programs and the self-responsibility to get regularly trained are important to provide safe and evidence-based care for patients. HCPs, as well as nursing students, in Germany are not explicitly required to attend special patient safety compliance and improvement trainings on a regular basis. Consequently, it is not ensured that HCPs are trained in topics such as speaking up, using critical incident reporting systems, standardized handovers in all clinical areas and can work safely. This aspect is the subject of numerous health policy debates to improve education and training in nursing and medical fields [41,42]. This underlines the importance of a safety culture and safety performance in acute care once more. Empowerment training for nurses that aimed to improve safety culture was found to significantly impact the clinical practice [43,44]. It focused especially on communication domains like openness, speaking up and error communication [43], aspects which were mentioned as important but also inadequate at the frontline in this study.

### Limitations

There are several limitations to our research that should be considered when interpreting the results of our study. Social desirability bias may have affected our results. The topic of safety performance in acute care can be particularly influenced by social desirability, and consequently the interviewees may not have spoken openly about sensitive events, such as errors in acute care settings. The study results’ generalization could be limited by the self-selection bias, as the volunteer participants may not be representative of the entire healthcare professionals. And a self-serving bias could also have influenced the response behavior and limited the interviewees’ ability to reflect on their performance and role as HCPs. Furthermore, our sample consists of registered nurses, nursing students and pedagogical personnel, so the interviews primarily reflect the perspectives of these professions. The special training as a registered nurse in Germany, the involvement of students from the start of training, and the involvement of pedagogical personnel in nursing teams must be taken into account while interpreting the results. Future studies should be based on heterogeneous samples so that the average of HCPs in Germany is represented. Authors should discuss the results and how they can be interpreted from the perspective of previous studies and of the working hypotheses. The findings and their implications should be discussed in the broadest context possible. Future research directions may also be highlighted.

## 5. Conclusions

This study aimed to examine HCPs’ perceptions about safety performance with a qualitative approach. Results indicate on the one hand that HCPs fail to have a more comprehensive and complex picture of safety performance at the frontline and, on the other hand, that organizational aspects have a huge impact on safety performance, and compliance to rules and standards. HCPs need regular trainings in safety performance and patient safety, provided by their organization. Based on these findings, HCPs working in nursing should be more aware of safety performance and patient safety to be more conscious of their role and have a deeper understanding of the interactions between individual, team, patient, organization, and the work environment. The necessary basic qualification of nurses should also be critically examined for Germany against the background of the international standard of higher education qualifications in nursing.

Further studies should focus on interventions to socialize nurses for patient safety and safety performance from the beginning of their education and explore inter-professional teams’ experiences to get a deeper understanding of safety performance at the frontline.

## Figures and Tables

**Figure 1 healthcare-09-01543-f001:**
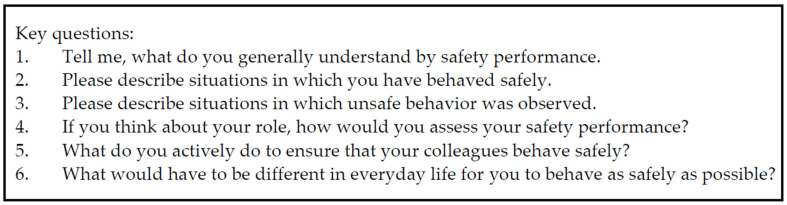
Key questions of the semi-structured interviews.

## Data Availability

The data presented in this study are available on request from the corresponding authors.

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
