# Peer review of "Safety Performance in Acute Medical Care: A Qualitative, Explorative Study on the Perspectives of Healthcare Professionals"

_healthcare, 2021, doi:10.3390/healthcare9111543_

Round 1

Reviewer 1 Report

I would like to congratulate the authors on what is, in my opinion, a very good manuscript. Please find my detailed comments in the annexed PDF file.

Author Response

Dear Reviewer

Thank you very much for taking the time and effort to review our manuscript, we appreciate your careful review and important recommendations. A revision of the paper has been carried out to take all of them into account. We responded point-by-point to your comments, which you can find below.

Kind regards,

The Authors

Overall, I believe this is a very good study. In fact, it serves as a good example of why qualitative research can provide valuable information, helping disprove the commonplace idea that quantitative research is not only superior but can stand on its own. The topic explored by the authors is, likewise, an undoubtedly important, often overlooked aspect of healthcare: safety.

I believe this work is an interesting exercise in understanding nurses’ perspectives on security. I have only a few remarks to make, regarding aspects the authors could address to improve their already well- written work. Please find them below:

Abstract

The Abstract is well written. It is clear and provides a good summary of the study. It provides an adequate overview of the work’s methodology and conclusions.

Introduction

The Introduction is detailed, without being repetitive, and provides an accessible overview of the background that this study emerged from. There is, however, a small detail I’d like to point out:

- The sentence starting on line 38 could be rephrased and made clearer. In fact, it is not evident whether the authors are referring to factors influencing nurses’ work or simply enumerating several aspects of it. Furthermore, when they write more actual ill patients, it is hard for me, as a reader, to understand what an actual ill patient is; the implication seems to be that there are patients that are actually ill and those that there are not, which does not make sense.

Methods

The Methods section provides a good exposition of the work conducted by the authors. The study’s setting and sample are particularly well described. I would like the authors to clarify the following:

- On line 113, the authors refer to a brainstorming session that was behind the elaboration of the semi-structured interview’s questions. In my opinion, it would add to the description if the researchers detailed and justified whether they opted to come up with questions based solely on their expertise or if these were based on previous literature or theoretical backgrounds. I understand the interview structure in itself was based on the CRSS method, but what about its content?

- Starting on line 139, it is described how the authors came up with a categorization system for the themes found in the content analysis. It is made clear that the categories were devised based on a theoretical model, which is an extremely valuable detail. I would like to ask, however, if there were taken any measures to ensure that the development of categories was unbiased, i.e, if category theory was used.

Results

The Results section is well-structured, and the authors provide a good picture of the data they gathered. I have no questions or further comments to make.

Discussion and Conclusions

The two last sections are well-thought-out, and the authors were invested in their work and were able to extract valuable insights from the collected data. I have only one remark:

- In the results section, starting on line 203, it is referred that HCP receive regular, formal safety training, even though the nature of this training is not entirely clear. I believe this is a dimension that could be explored further in the discussion/conclusions. In fact, the authors recommend, on line 369, that HCP should have regular safety training. Does this mean the training they currently have is not enough? It might be interesting for the authors to acknowledge the training that HCP are said to receive currently and compare it to what an ideal training would be like. As such, even though they have no data to give definite, clear recommendations for future training, I think that the authors could be more detailed and give their expert opinion on how training programs could improve, so that mentioned issues like HCP struggling to describe safety issues (line 305) or minorizing their role in detriment of the organization’s (line 319), for example, could be addressed

Comment

Response

Overall, I believe this is a very good study. In fact, it serves as a good example of why qualitative research can provide valuable information, helping disprove the commonplace idea that quantitative research is not only superior but can stand on its own. The topic explored by the authors is, likewise, an undoubtedly important, often overlooked aspect of healthcare: safety.

Thank you very much for your positive feedback about our work!

Abstract

The Abstract is well written. It is clear and provides a good summary of the study. It provides an adequate overview of the work’s methodology and conclusions.

Thank you very much for your positive feedback.

Introduction

The sentence starting on line 38 could be rephrased and made clearer. In fact, it is not evident whether the authors are referring to factors influencing nurses’ work or simply enumerating several aspects of it. Furthermore, when they write more actual ill patients, it is hard for me, as a reader, to understand what an actual ill patient is; the implication seems to be that there are patients that are actually ill and those that there are not, which does not make sense.

Thank you very much for pointing this out. We have clarified and revised the sentence. Please see lines: 39-41.

Methods

On line 113, the authors refer to a brainstorming session that was behind the elaboration of the semi-structured interview’s questions. In my opinion, it would add to the description if the researchers detailed and justified whether they opted to come up with questions based solely on their expertise or if these were based on previous literature or theoretical backgrounds. I understand the interview structure in itself was based on the CRSS method, but what about its content?

Thank you for your comment. We have explained the content of the CRSS method we used for development of the interview guideline in more detail. Please see lines: 121-124.

Methods

Starting on line 139, it is described how the authors came up with a categorization system for the themes found in the content analysis. It is made clear that the categories were devised based on a theoretical model, which is an extremely valuable detail. I would like to ask, however, if there were taken any measures to ensure that the development of categories was unbiased, i.e, if category theory was used.

Thanks for the question, we are not sure what is meant by a "category theory".

Our approach was a theoretical framework model, coding and interpretation with multiple researchers, interdisciplinarity, reflexivity in terms of one's perspective on the content ensured in the team, and following the standards for reporting qualitative research (SRQR). The view of the researchers and his/her characteristics and reflexivity was always being part of the qualitative research process and was constantly reflected and described in the process of data collection and data analysis. Thus, we do not believe that our coding and category formation can be unbiased.

Results

The Results section is well-structured, and the authors provide a good picture of the data they gathered. I have no questions or further comments to make.

Thank you very much for your positive feedback.

Discussion and Conclusions

In the results section, starting on line 203, it is referred that HCP receive regular, formal safety training, even though the nature of this training is not entirely clear. I believe this is a dimension that could be explored further in the discussion/conclusions. In fact, the authors recommend, on line 369, that HCP should have regular safety training. Does this mean the training they currently have is not enough? It might be interesting for the authors to acknowledge the training that HCP are said to receive currently and compare it to what an ideal training would be like. As such, even though they have no data to give definite, clear recommendations for future training, I think that the authors could be more detailed and give their expert opinion on how training programs could improve, so that mentioned issues like HCP struggling to describe safety issues (line 305) or minorizing their role in detriment of the organization’s (line 319), for example, could be addressed.

Thank you for your suggestion to clarify the training aspect of safety performance in more detail. We added a section to discuss training for patient safety and its effect for safety performance. Please see lines: 350-355.

Reviewer 2 Report

The article addresses the important issue of patient safety with the purpose to get a deeper understanding of healthcare professionals' perspective in acute care on the topic of safety performance.
The data collection methods comprised qualitative interviews and a cross-sectional written survey.
At the end of paragraph 2.1 the results of the first SPOHC survey should be shortly mentioned.
The most important limitation of the study is that the sample is very small, so I would recommend emphasizing in both the title and the study that this is a preliminary study.
In the discussion, the authors should give some practical examples of how to implement safety levels (Handover, incident reporting, patient safety walkaround). There is a lot of literature and these articles might be useful: doi: 10.3390 / ijerph17176267; doi: 10.2147 / RMHP.S129652.
In paragraph 4.1 the limitation due to the low number of the sample under study should be added.

Author Response

Dear Reviewer

Thank you very much for taking the time and effort to review our manuscript, we appreciate your careful review and important recommendations. A revision of the paper has been carried out to take all of them into account. We responded point-by-point to your comments, which you can find below.

Kind regards,

The Authors

The article addresses the important issue of patient safety with the purpose to get a deeper understanding of healthcare professionals' perspective in acute care on the topic of safety performance.
The data collection methods comprised qualitative interviews and a cross-sectional written survey.
At the end of paragraph 2.1 the results of the first SPOHC survey should be shortly mentioned.
The most important limitation of the study is that the sample is very small, so I would recommend emphasizing in both the title and the study that this is a preliminary study.
In the discussion, the authors should give some practical examples of how to implement safety levels (Handover, incident reporting, patient safety walkaround). There is a lot of literature and these articles might be useful: doi: 10.3390 / ijerph17176267; doi: 10.2147 / RMHP.S129652.
In paragraph 4.1 the limitation due to the low number of the sample under study should be added.

Comment

Response

The data collection methods comprised qualitative interviews and a cross-sectional written survey. At the end of paragraph 2.1 the results of the first SPOHC survey should be shortly mentioned.

Thank you very much for your suggestion. We have added a sentence and mentioned shortly the main results of the survey. Please see lines: 76-80

The most important limitation of the study is that the sample is very small, so I would recommend emphasizing in both the title and the study that this is a preliminary study.

Thank you very much for your comment. We have adapted the title of the manuscript and the description of the mixed-methods study and explicitly listed that SPOHC is an exploratory study. Please see the title and line: 70

In the discussion, the authors should give some practical examples of how to implement safety levels (Handover, incident reporting, patient safety walkaround). There is a lot of literature and these articles might be useful: doi: 10.3390 / ijerph17176267; doi: 10.2147 /RMHP.S129652.

Thank you for pointing this out. We have added your helpful literature suggestions and gave some practical examples. Please see lines: 330-331.

In paragraph 4.1 the limitation due to the low number of the sample under study should be added.

Thank you very much for this comment. We conducted 23 interviews and conducted them to a point where no new categories and codes emerged. In our view, the number of interviews is not a quality criterion of qualitative research (Guest G, Bunce A, Johnson L. How many interviews are enough? an experiment with data saturation and variability. Field Methods. 2006; 18: 59–82). A limitation of the validity is the fact that sampling does not allow comparative or contrasting analyses between occupational groups. We have included this point in the discussion (see lines 370-372; 374-378). However, this was not an aim of our study. The goal was to explore the perspective of healthcare professionals in general on the subject matter. This was achieved with the number of interviews and the intentional sampling.